# Local adaptation and coping strategies to global environmental changes: Portraying agroecology beyond production functions in southwestern Ethiopia

Gemechu Y. Ofgeha[1,2]*, Muluneh W. Abshare[1]

1 Department of Geography and Environmental Studies, Addis Ababa University, Addis Ababa, Ethiopia,
2 Department of Geography and Environmental Studies, Wollega University, Gimbi, Ethiopia

* dandigemechu11@gmail.com

**Data Availability Statement:** All relevant data are within the manuscript and its S1, S2 Texts files.

**Funding:** This work was supported by Addis Ababa University, Ethiopia.

## Abstract

### Background

The recent research recommendations on the adaptations of poor are toward local specific investigations, aimed at a comprehensive understanding of the adaptation strategies through in-depth analysis of the status, and the explicit on how climate and non-climate global change processes constrain the inherent strategies. Intent to this idea, we have designed this study to assess the small-scale farmers' adaptation and coping strategies in southwestern Ethiopia.

### Methods

The agroecology approach steered in case-study design was used for the conceptual and analytical framework. The data collected from 335 households were analyzed for descriptive and multivariate analysis of variance and substantiated by qualitative data obtained through focused group discussion, interview, and observations.

### Results

The significant differences were observed in the watershed among households in the case studies on their adoption of the identified adaptation and coping strategies. The sustainability of preferred strategies was different along case studies, solely determined by the impact magnitude of the adaptations constraining factors. Although free ecosystem-based strategies become less practical and replacing by new strategies in the watershed, the processes were gradual, internal to the community and managed through adaptive learning in the highland. However, the paths were perceived as toward maladaptive, resulted by the state interventions which disrupted free adaptations, deteriorated adaptive learning of the community, and shaped the adaptation responses toward the interventions in the *kolla* agroecology.

**Competing interests:** We declare that there are no conflicts of interest.

## Conclusions

The study implies that the situations of households' adaptation strategies are beyond the reflections of their respective production ecology, designated within climate variability in the previous studies. The structural land use dynamics and associated resource tenure insecurity have greater constraining effects on the strategies than the impacts of climate variability in the *kolla*. Thus, subsequent research interested in such contexts, and any plan for the development interventions should (re)consider the impacts of non-climate national/and global environmental change in shaping the adaptation and coping strategies of the local community.

## Introduction

The researches on the local livelihood of the poor in the developing world come up with the complexity of adaptation constraints due to highly dynamic and challenging impacts of global environmental changes of climatic and non-climatic site-specific factors [1, 2]. For instance, the synthesis of research on barriers of adaptation conducted on sub-Saharan African countries is typified by persistent poverty and socio-economic inequality, low levels of development, high dependence on climate-sensitive livelihood sectors, limited economic capacity, and numerous governance and institutional challenges on top of the impacts of climate change, resulting in low adaptive capacity and significant adaptation deficit [3, 4]; most importantly forced the community to mal-adaptation strategies [5].

Although agroecology has conceptualized in line with production functions in most of the studies conducted in Ethiopia, in this study we argued that the conceptualization should be beyond the ecological function mainly when applied for the local livelihood assessments. Particularly, the adaptation and coping strategies of small-scale farmers in the Anger watershed couldn't be understood by labeling agroecology zones as the production ecology only, fixing constraining factors to climate and biophysical features as far as the environment and livelihood situations of the community are concerned [6], but considering the site-specific non-climate national/ global change processes has paramount importance. Specifically, climate variability and structural land-use dynamics were identified and defined as global change processes external to the community and the ecosystems through observations of the contexts in the study area and critical review of previous studies [3, 6–8].

For clarity, the identified processes of climate variability and structural land-use dynamics have global faces and impudent in the study area with distinguished spatiotemporal settings due to differences in climate state and variability situations across agro-climate in one hand and the historical socio-economic and political spurs of agroecology resulted mainly from national/ and global interventions on the other hand. To this end, such contextualization is crucial to understand the conditions of farmers' adaptation strategies, specifically, the status, trends, causal linkage of the constraints to adaptation strategies along with the processes' contexts in the study area [6].

Moreover, the recent recommendations are toward a much emphasis on the assessment of agroecological aspects of non-climate processes, and systematically incorporating the impacts of these processes with the climate variability to understand local situations of adaptation strategies [9, 10]. Therefore, an in-depth investigation of the agroecological aspects of each change process separately and their combined impacts on the adaptation strategies of the local community was the central purpose of this study. These issues become among the insightful

concerns of today's human dimension of global environmental change research [11]. However, these non-climatic contexts are seldom considered explicitly in identifying the spatio-temporal difference in their impacts on the adaptation strategies. Particularly, why adaptation strategies experienced by the community are sustainably effective in some specific local areas, but not in other places, what factors are significantly constraining the strategies, and how these constraints are associated with local specific environmental change processes.

The researches on the adaptation and coping strategies of the local community to climate change in Ethiopia were conducted at a large spatial scale, and a significant attention wasn't given to the unique characteristics of the local area in the assessments of the strategies [12–15]. Most of these researches used natural division such as agroecology for the comparative analysis [16–18]; and climate change was the only stressor considered in these studies. However, there are substantive roles of spatially different biophysical, socio-economic, and historical distinctiveness shaping the local adaptation strategies through the effects on the local community's resource endowments, entitlements, and adaptation capabilities. Thus, research on this issue, as settled on these specified gaps is crucial in the Anger watershed. These are toward identifying adaptation and coping strategies of small-scale farmers' households at the local level, examining the difference in the household's adoptions along with the case studies, and understanding the causal linkage of adaptation constraints with site-specific stressors. These have paramount importance provide new insight in adaptation research and development policy recommendations. These could be achieved through understanding the complex interactions of the processes, and the synergetic impacts of these local-specific environmental changes on the adaptation responses of the households along with agroecological segments of the community.

To this end, the lack of access to land resources due to structural land-use dynamics and climate variability were the stressors observed for their effects on the small-scale household's adaptation in the study area [6]. The community has been responding to these environmental, socio-economic and political hurdles they are facing by their efforts through the local adaptation and coping strategies. These external disturbances have spatiotemporal differences in the study area. Consequently, where the magnitude of the impacts are acute the local community's adaptation responses to the stressors, and the sustainability of their inherent strategies are adversely affected. Therefore, the status and trends of the local adaptation strategies are the reflections of these local specific environmental change processes, mainly for their current practicality, effectiveness, and sustainability.

In this study, the structural land-use dynamics is the changes in ownership of land and its property as a result of resource use governance (land tenure system), the changes in access to the resources among the local community, the state, and large-scale agricultural investors driven by national land-use policy changes [19], and globalization- mainly with the recent increase in the global foods and energy demands [20, 21] unlike commonly observed drivers of the land use and land cover change in Ethiopia. These change processes have unique characteristics defined by the economic systems and land policies of the time for their environmental and livelihood impacts and end with system and policy changes [6]. Thus, the issues of the local adaptation have been conceptualized in this study with the strong assumption that these processes have differential implications on farmers' adaptation situations along with the spatio-temporal patterns of these processes.

The general objective of this study was to appraise the local adaptation and coping strategies of the households by considering the agroecological contexts in the Anger watershed. The specific objectives of this study were as follows: (1) to assess the status and trends of household's adoption of the local adaptation and coping strategies. (2) to examine the spatial variations in the household's adoption of the identified strategies in the case studies. (3) to explore context-

specific causalities of adaptation constraints. The working hypotheses were: 1) there is no difference in household's adoption on the linear combination of adaptation and coping strategies based on their difference in case locations, 2) the *kolla* and highland farmers do not significantly differ in their adoption of the adaptation strategies, 3) the *kolla* and highland farmers do not significantly differ in their adoption of the coping strategies.

## Conceptual frameworks

Ontologically, human endeavours to adapt/cope with the impacts of global environmental changes for survival is a permanent reality has existed since antiquity. The concept of adaptation was in the debates about resource scarcity in the early development researches [1]. But recent studies on livelihood adaptation strategies have increasingly been used in different contexts. For instance, the adaptation to the anthropogenic climate change [22], global economic changes [23], socio-political systems disruption by exposure to a civil war [4, 24] and many other context-specific studies have been taking place with different objectives and frameworks. These extensive uses have contributed to capturing different contexts in this study through integrating these conceptual frameworks. The issues and problems in this study were context-specific, which consist of various global environmental change issues such as climate variability and global economic changes. Thus, these conceptual frameworks are integrated into this study to understand how the processes at the local agroecology have been affecting household's adaptation strategies in the study area. The local adaptation impacts of global change process such as climate change, globalization, regional and local 'developments' [23] and how the processes exhibited in adaptation response of local farmers [8, 25, 26]. These integrations were enabled us to identify how the local adaptation strategies are affected by the access to resources, entitlements, livelihood capabilities and power principles [26] among poor small scale farmers in the phase of changing climate. These were done through identifying trends in adaptation strategies, status on who adopt which strategy, and why; most importantly to imply constraints to adaptations, and livelihood pathways along with case studies. These have been rarely done while omission of these local contexts leads to critical weaknesses in understanding causal-linkages of adoptions constraints [2].

Adaptation is an adjustment to behaviour or economic structures that reduce the vulnerability of society in the face of scarcity or threatening environmental change [26]. This definition is comprehensive which included response/and adjustment to any type of stressors typically observed in a specific area. The local-specific non-climate stressors have been superimposed upon the impacts of climate change to shape the adaptation strategies and adaptive response of farmers in Ethiopia [17]. Therefore, the local adaptation situations and the status of strategies have complex spatiotemporal dimensions due to the difference in the area extent and impact magnitudes of these non-climate stressor/s on top of the impacts of climate variability which varies across agroclimatic.

The study was framed in the concepts of livelihood resilience to double exposure [23], integrated with agroecological approach [27] to fit the central argument of the study. The double exposure framework captures the rapid, on-going and local context climatic and non-climatic processes, and the impacts in complicating households' adaptation. The agroecology approach in this case is socioecological constructions, the simultaneous applications of both production and political ecology. Thus, adaptation situations across the agroecology in the study area were not resulted by the biophysical dynamics and the associated resources and production functions of the ecology, but the power relations as well. There is a close relationship between agroecology and biophysical dynamics such as climate change on one hand, and the national/and global economic and political dynamics in the watershed. Therefore, the simultaneous impacts of these dynamics play a crucial role in the adaptation strategies and options of the local

community [28]. Particularly, the adaptation and coping strategies among the small-scale farmers in the Anger watershed couldn't be understood by labeling agroecology zones as production ecology, fixing constraining factors to climate and biophysical factors as far as the environmental and livelihood systems in the community are concerned [6].

Thus, we proposed the concept of agroecology of adaptation strategies for this study, hypothesizing the conditions for its applicability beyond the conception commonly applied in the previous studies. Explicitly, the agroecological-based difference in the biophysical/and climate processes and internal socioeconomic and demographic vitalities of a community have been used as central issues for the assessments of sustainability and constraints to local adaptation strategies. Acknowledging these, we claim that non-climate global environmental changes are a reality in the study area, the process has agroecological aspects, and the recommendations for considering such processes are crucial for the study. Specifically, through observations of the contexts in the study area and critical review of previous studies, the climate variability and structural land-use dynamics have been identified and defined as the global environmental changes, and these processes are external to the community and physical environment in the study area [6–8].

This adopted conceptual framework was followed by the case studies-based analysis of the adaptation situations. In the context of the watershed, the global environmental change processes vary from place to place. These spatial variations in these processes resulted in the impact difference on adaptation situations and in constraining the strategies. To this end, in line with the studies conducted by these approaches [29–32], we have decided on a case-by-case analysis. The case studies were designed based on these contexts.

The studies on the local adaptation strategies are approached differently, which are broadly associated with the disciplinary orientations, system to be studied, the level of analyses, and the objectives of the study. Taking these into consideration, the development approaches to adaptation studies [33, 34] were used in this study. This approach provides to assess adaptation strategies at the household level, consider non-monitory measures, and integrate non-climate threats in the analysis. The small-scale farmers' adaptation at the local scale with the objective of assessing the difference in the status of adaptation strategies and the causal linkages of adaptation constraints were the central issues in this paper to be addressed by this approach. Moreover, both deductive and inductive approaches were used to identify the variables and collect data for this study. The researches on farmers' adaptation in Ethiopia were reviewed and integrated with the discussions with the experts and the local community during preliminary fieldwork were used in this study. Thus, the exhaustive lists of the strategies were summarized into eleven adaptations and seven coping strategies practiced in the study area. These strategies were separately used as adaptation and coping in this study due to the fact that the adoptions of the strategies were different among the households and along with the case studies which were explicated during preliminary fieldwork on top of previous empirical justifications. Accordingly, based on the duration of the stressors, copings are the strategy to the short-term shocks, and adaptation is for the long-term impacts [16, 29] of climate variability and non-climate factors. Finally, we hypothesized that the situations of the local adaptation in the watershed is resulted by the synergetic impacts of these processes, that the differences along the agroecology on the sustainability of the strategies and their constraints were due to the spatial variations in the distributions of these stressors, and the associated differential effects at the local level [6].

## Research methods and materials

### Case studies

Anger watershed lies in east Wallagga administrative zone extended from 09°12′ to 10°00′ north and 36°30′ to 37°11′ east was the focus of the study, while the watershed covers other

administrative zones of Oromia regional state and regional state of Benishangul Gumuz. The selected single administrative zone was to reduce heterogeneity in socio-economic, cultural, and livelihood systems and differences resulted from regional and local governance on the issues of this study. The watershed is composed of highlands(3.4%), midlands(39.6%), and lowlands (56.1%) with altitude ranges from 1200 to 3018 meters above sea level [6]. The variations in relief and topographic arrangements resulted in differences in climate and distribution of natural resources such as soil types, drainage patterns and water resources, and forest ecology. The socio-economic conditions of the watershed, such as population distribution, land use pattern and potentials, and economic activities were shaped by these biophysical on tops of historical and political factors.

Two case units were identified in line with previous vulnerability study in the watershed [6] for the comparative analysis of persistent adaptation and coping strategies along with the case units, how the strategies are shaped by the local specific factors, and to associate the adaptation constraints with the underlying situations. Each case unit has its biophysical, socio-economic, and historical characteristics. These differences are believed to shape the small-scale households' adaptation and coping strategies. Accordingly, the households in the *kolla* were the first case unit characterized by tropical/*kolla* agroecological conditions and experienced structural land-use dynamics. The second case unit lies within sub-tropical/*woinadega* and temperate/*dega* agroecology and both are out of the influence of historical structural land-use dynamics.

The small-scale farmers in these case studies have a significant difference in their socio-economic contexts such as land size for agriculture, access to grazing land, water sources, and other land resources. These resources provide the livelihood assets for the local community, but the tenure conditions on the local resources and the extent to which the community depends on these ecological resources vary along with the cases. Moreover, the climate and biophysical situations have the difference. These differences have resulted from the agroecological positions in one way or the others. Thus, understanding the spatio-temporal dynamics of the local adaptation and coping strategies is crucial to identify who adopts which mechanism, the significance of adoption difference, and the causal linkage of the adaptation constraints. Therefore, the data collected from various sources through different techniques were analysed to examine the difference in adoptions along with case studies and to construct the causality; i.e., the extent to which these global environmental change interventions caused the results particularly, the perceived adaptation constraints and the local response to these stressors.

## Approach and design

This study consists of an array of issues which ranges from examining the local adaptation and coping strategies along with case-studies by using the statistical models to constructing the causalities for the adaptation constraints through in-depth exploration of the local concerns. In line with these, the mixed research method approach was used [35]. Specifically, the narrative inquiry was integrated with and embedded in the model-based examinations to understand the complex social-ecological system, particularly the interactive, systemic, and constantly emergent nature of the local community's adaptation situations. This approach was applied to collect a list of facts, a progression of events connected to ways the study participants learn, explain, and organize their experience on the issues [36] to capture the processes. Thus, the narratives on the local situations of the environmental change processes were interpreted critically and integrated with the local community's historical accounts on the adaptation and coping strategies.

The case study research design was used in this study. First, the case unit was defined as the area of the Anger watershed characterized by uniform conditions regarding the adaptation situation in the contexts of this study. Moreover, the case study design in this study was the

comprehensive research strategy for intensive analysis of an individual case unit having a unique feature rather than methodological choice [37]. Therefore, a case unit may be studied in different strategies. For instance, qualitatively or quantitatively, or mixed methods. The methodological choice is not decisive for whether it is a case study or not, rather the demarcation of the unit's boundaries according to sets of unique and uniform features was used to design the case study in this study. Therefore, the investigation and presentation were done through both descriptively and exploratory depending on the nature of the specific objectives. For instance, the status of adaptation and coping strategies and comparisons on the adoptions of the strategies along with case-studies were analysed by descriptive and inferential statistics respectively. The analysis on the causal linkage of adaptation constraints was guided by exploratory design. The issue on the why and how adaptation and coping strategies of the local community have been shaped by case-specific situations was discovered through an in-depth examination of the community's perception.

## Research methods

The small-scale farmers' household heads, and the focus group discussants drawn from local notoriety individuals and the elderly in the Anger watershed, and experts from relevant zonal and districts offices in the east Wallagga administrative zone were the primary sources of data. The published and unpublished documents were used as secondary sources in this study. Both quantitative and qualitative data were collected through questionnaires, focused group discussions (FGD), discussion with experts, and observations.

The multistage sampling technique was used in this study. The areas found within the Anger watershed in the east Wallagga administrative zone were the focus of the study. First, small-scale farmers' household heads were purposively selected as the unit of analysis. Second, the divisions within the region were made on the basis of the agroecology and situation of structural land used dynamics. The study area entertains of three agroecology: the *kolla* agroecology within the area affected by the impacts of structural land use was identified for the first case unit. This area was characterized by the warm lowland climate having an altitude less than 1500m a.m.s.l situated in the central part of the watershed The second case unit consists of the midland (*woinadega*) and the cool temperate (*dega*) agroecology in the midland and mountainous areas of the watershed. Then, districts and kebeles that represent both cases were randomly selected, from which 335 sampled households were identified using the following formula [38]. In deciding sample size, all necessary criteria [39] were considered. The sample size for each kebele was determined based on the proportionality of the size in the household's population. Finally, the simple random sampling technique was used to select household heads for the questionnaire survey.

$$n = \frac{z^2.p.q.N}{e^2(N-1) + z^2.p.q}$$

Moreover, the participants on FGD were purposively selected. Four FGD group having members of eight individuals for each group were arranged (i.e., one group for each kebele).

Data analysis was performed using by the STATA for the descriptive and inferential analysis. The descriptive method was used to assess the status of adaptations and coping strategies practiced by the households. The multivariate analysis of variance (MANOVA) was applied to test the hypotheses of the study.

**Model specification for one way-MANOVA.** The data comprised categorical independent variables (*kolla* and highland), two classes of dependent variables (adaptation and coping strategies) and individual dependent variable (11 under adaptation and 7 under coping class).

The statistical model for a case design was:

$$Yij = \mu + \beta j + \varepsilon ij \tag{1}$$

Where; $Yij$ is $i^{th}$ observation in the j group; $\mu$ is the mean for all observations; $\beta j$ is the treatment effects in $j^{th}$ group; and $\varepsilon ij$ is the error term.

The distinctive role of the model is to combines these multiple dependent measures into a single value that maximizes the differences across the groups. It provides additional insights into the implications of independent variables (being in the *kolla* or highland) on dependent variables (adoptions of the strategies) [40].

MANOVA has several statistical assumptions by which three main assumptions are cautioned in several researches [41]. First, the normality, which referring to the shape of the data distribution and its correspondence to the normal distribution should meet. Second, the covariance matrices for the treatments must be equal. Third, the observations must be independent. The households survey was individually administered and observed independently. Moreover, for the significance tests, Wilks' Lambda is recommended for general use, but the Pillai's Trace was used due to unequal households size in the *kebeles* for the case studies [42] to test the null hypothesis.

$$H_o = \begin{pmatrix} \mu 11 \\ \mu 21 \end{pmatrix} = \begin{pmatrix} \mu 12 \\ \mu 22 \end{pmatrix} = \ldots\ldots\ldots = \begin{pmatrix} \mu 1g \\ \mu 2g \end{pmatrix} \tag{2}$$

Where; $\mu_{ig}$ ($i$ = 1 and 2) is the population mean for the strategies in the *kolla* and highland respectively for both $g$; adaptation and coping group. To test the differences between the groups i; mean vectors of the $g$ (11 for adaptation and 7 for coping); the significance test is specified as:

$$\Lambda = \frac{|S_{error}|}{|S_{effect} + S_{error}|} \tag{3}$$

The significant test statistic for MANOVA works closely with the F-test [40]. An estimate of *F* was calculated through the following equation:

$$F_{approximate} = \left(\frac{1 - \sqrt{\Lambda}}{\sqrt{\Lambda}}\right)\left(\frac{n - g - 1}{g - 1}\right) \tag{4}$$

Where; n = sample size; and $g$ is group of strategies.

Moreover, the data collected through the focus group discussions (FGD) and field notes were qualitatively analysed to explore the similarities and differences in the situations of adaptation and coping strategies across the case studies. This technique was applied to identify how and why a certain strategy has been sustainably practiced in a given area and not in other places. These were mainly to pinpoints the causal linkages of adaptation constraints. During the FGDs, the participants were facilitated to storytelling and to discuss their experiences in adaptations.

## Ethical consideration

The ethical issues related to research subjects which broadly manifests in advocacy, anonymity, confidentiality, voluntary and informed consent have due consideration in this study. The research protocol was reviewed and approved by Wollega University Research Ethics Approval Committee of College of Social Sciences, Gimbi (Reference Number: WUGC 0015/D-18/12/

2019). The letter issued by the committee was used to request permissions from concerned bodies in the study area before the commencements of data collection. Thus, the zone and districts administrations in the study area have approved to get access to the study site and participants. The focus of informed consent was realized during data collections through discussions with the participants on the purpose of the research, duties, and responsibilities of and participants. The letter was submitted to the participants and verbal consent was requested by reading the letter for those respondents cannot read the letter. All the participants were above 18 years that the written consent was directly obtained from them in the form of a signature and a thumbprint. Confidentiality and privacy of the respondents have been realized that the survey data and field notes were kept confidential.

## Results

### Status of adaptation and coping strategies

Both descriptive and multivariate analyses of variance were applied to compare adaptation and coping strategies along with case studies. The descriptive analysis (Table 1) was to identify the status of the identified types of adaptation and coping strategies practiced by the households across case studies.

All identified adaptation and coping strategies have been practiced by the local households in the watershed, while the extents of adoptions for each differ within and along with the case studies (Table 1). Except for reducing the stocking density that shows an equivalent adoption along with the case studies, all adaptation strategies show remarkable differences. The applications of soil fertility management, changes in planting date, crop diversification, uses of improved seed, small scale irrigations, practices of SWC were the most applied strategies in the highland. The engagements on non-farm activities and reducing stocking density, practicing tree planting were the most practiced adaptation strategies in the kolla. These results indicated that small-scale farmers in the highland have more engaged in adaptation strategies than farmers in the kolla, where non-farm activities were the most practiced strategy.

On the other hand, although the level of adoption was much lesser than adaptation strategies, several coping strategies have been practiced in the watershed. Accessing credit was the most practice strategy in both case studies but slightly more adopted in the highland. Among coping strategies, seasonal migration, remittance, and selling livestock have been more practical by the households in the kolla, while the extents of coping strategies through remittances

**Table 1. Extents of farmers' adoptions of adaptation and coping strategies across case studies.**

| S. N | Adaptation mechanisms | *Kolla* (%) | High land (%) | Coping mechanisms | *Kolla* (%) | High land (%) |
|------|----------------------|-------------|---------------|-------------------|-------------|---------------|
| 1 | Change in planting date | 2.2 | 79.6 | Change in consumption | 7.9 | 10.2 |
| 2 | Crop diversification | 41.0 | 77.6 | Receiving support (PSNP) | 8.6 | 14.8 |
| 3 | Using improved seed | 28.8 | 77.0 | Accessing credit | 51.8 | 65.8 |
| 4 | Engaging on non-farm activities | 80.6 | 21.4 | Selling livestock | 20.9 | 7.1 |
| 5 | Reducing stocking density | 61.2 | 60.2 | Renting out land | 5.8 | 20.0 |
| 6 | Improving livestock farm | 26.6 | 19.9 | Seasonal migration | 46.8 | 15.3 |
| 7 | Employing small scale irrigation | 25.9 | 75.0 | Remittance | 32.4 | 24.0 |
| 8 | Practicing tree planting | 61.9 | 54.1 | | | |
| 9 | Practicing SWC | 13.7 | 74.5 | | | |
| 10 | Applying soil fertility management | 38.8 | 81.1 | | | |
| 11 | Using drought-tolerant crops | 12.2 | 11.2 | | | |

Source: Household survey, 2020

and renting outlands were also remarkable in the highland. Contrary to the highland, where adaptation strategies were substantial, the results revealed that the small-scale farmers in the kolla were relatively higher in adopting most of the coping strategies (Table 1).

## Comparative analysis of local adaptation and coping strategies

The MANOVA output presented below (Table 2) describe the difference among strategies on the dependent variate (i.e., whether or not they vary along with the cases, and the difference could produce a significant multivariate effect on the adoptions). In addition, it assumes a cause-effect relationship whereby the groups (independent variables) and controlled variables (adaptation and coping classes) cause a significant difference in the adoption of an individual strategy.

The statistical assumptions MANOVA such as missing values, outliers, linearity, normality, and homogeneity of the variance matrices were checked during data screening and presentations. Multivariate outliers were checked by computing a Mahalanobis distance measure for each strategy that no outlier observed in the model. The normality tests (the value of Mahalanobis distance) were higher than the critical value from the chi-square table, indicating that the violations of normality were not present in the dependent variables (Table 2[a]). Bartlett's test of sphericity was statistically significant for both adaptation and coping class (Table 2[b]), which indicates the sufficient correlation between the dependent variables.

The multivariate tests of significance indicated that there are statistically significant differences between the case studies on the adoption of both adaptation and coping strategies. The

**Table 2. MANOVA output for the analysis of household's adoptions of the adaptation and coping strategies along with case studies in the Anger watershed.**

| Adaptation mechanisms | F-value | Tests of Between-Subjects Effects | | Coping mechanisms | F-value | Tests of Between-Subjects Effects | |
|---|---|---|---|---|---|---|---|
| | | Partial Eta Squared | Sign. | | | Partial Eta Squared | Sign. |
| Change in planting date | 593.79 | 0.914 | 0.000 | Change in consumption | 2.750 | 0.01 | 0.106 |
| Crop diversification | 137.73 | 0.864 | 0.000 | Receiving support | 5.278 | 0.02 | 0.020 |
| Using improved seed | 162.15 | 0.886 | 0.000 | Accessing credit | 7.616 | 0.02 | 0.01 |
| Non-farm activities | 248.52 | 0.854 | 0.000 | Selling livestock | 2.427 | 0.01 | 0.125 |
| Reduce stocking density | .212 | 0.848 | 0.652 | Renting out land | .012 | 0.00 | 0.912 |
| Improving livestock farm | 4.349 | 0.839 | 0.045 | Seasonal migration | 68.40 | 0.17 | 0.00 |
| Small scale irrigation | 135.06 | 0.852 | 0.000 | Remittance | 1.606 | 0.00 | 0.261 |
| Practicing tree planting | .184 | 0.888 | 0.672 | | | | |
| Practicing SWC | 228.05 | 0.857 | 0.012 | | | | |
| Soil fertility management | 125.58 | 0.867 | 0.014 | | | | |
| Drought-tolerant crops | .108 | 0.762 | 0.745 | | | | |

| [a] Residuals Statistics: Mahal. Distance Max = 33.260 | | | | Residuals Statistics: Mahal. Distance Max = 29.48 | | | |
|---|---|---|---|---|---|---|---|
| [b] Test of Equality of covariance Matrices: | Box's M | 266.2010 | | Test of Equality of Covariance Matrices: | | Box's M | 151.826 |
| | F | 3.8890 | | | | F | 5.297 |
| | df1 | 66 | | | | df1 | 28.000 |
| | df2 | 283096.6310 | | | | df2 | 308188.333 |
| | Sig. | 0.00 | | | | Sig. | 0.00 |
| [c]Multivariate Tests: Pillai's Trace (F = 136.180, P = 0.00; Partial Eta Squared = 0.82) | | | | Multivariate Tests: Pillai's Trace (F = 13.252, P = 0.00; Partial Eta Squared = 0.22) | | | |
| [d] Estimated Marginal Means: highland>*kolla* except for reducing stocking density | | | | Estimated Marginal Means: highland>*kolla* for change in consumptions and renting outland | | | |

Source: Household survey, 2020.

Pillai's Trace was used due to the unequal sample size for the case studies. The F-tests for Wilks' Lambda, Hotelling's Trace, and Pillai's Trace are identical in our multivariate test output. The multivariate main effect of adaptation strategies evaluated at hypothesis (between kolla and highland) show statistically significant (p = 0.00) differences between the case studies on the adoption of adaptation strategies. Moreover, the partial eta-squared value shows the main effect accounts for about 82.3% of the total variance between the case studies. Similarly, the multivariate main effect of coping strategies adoption produced a statistically significant effect (p = 0.00), indicating the difference between the case studies on the strategies. The partial eta-squared value shows the main effect accounts for only about 22.1% of the total variance between the case studies (Table 2ᶜ).

The statistically significant multivariate effect informs us that both adaptation and coping strategies are associated with differences between small-scale farmers on the level of adoptions along with case studies. This calls us, in turn, to presume that adoption difference exists and the next question is to discover which specific strategies are significantly different along with the case studies. Thus, the tests of between-subject effects (Table 2) output were used to answer this question. In this case, we have looked at a number of separate analyses of the variables, that a higher alpha level is expected to reduce the chance of a Type 1 error (i.e., finding a significant result when there isn't the real one). To this end, first Levene's test of equality of error variances was checked (the variable with p< 0.05 violated the assumption of the equality of variance for that variable). Accordingly, two variables of adaptation and four variables of coping violate the assumption. Thus, Bonferroni adjustment was done which involves dividing the original alpha level of 0.05 by the number of analyses (i.e., 0.05/2 = 0.025) [42]. Thus, the variables with p>0.025 were considered.

Household's adoption of adaptation was significantly different along with case studies in four strategies, namely: reducing stocking density, improving livestock farms, practicing tree planting, using drought-tolerant crops. These differences across the cases were significant again in four coping strategies, namely: change in consumption, selling livestock, renting out land, and remittance (Table 2).

The effect of being either in the *kolla* or highlands on the household's adoption of these adaptation and coping strategies can be evaluated using the effect size statistic provided in the Partial Eta Squared. It represents the proportion of the variance in the dependent variables (adaptation and coping classes) that explained by the independent variable (case studies: kolla and highland). These four significantly different adaptation strategies have Partial Eta Squared values that range from 0.763 to 0.841. These values, for instance, 0.841 represent that about 84.1% of the variance in reducing stocking density as an adaptation strategy explained by being in the kolla or highland among small-scale farmers, which, according to generally accepted criteria [40], was the very high effect. In the same way, significantly different coping strategies along case studies have Partial Eta Squared values ranges from 0.001 to 0.69, imply very low to medium effects on the variance in adoption along with case-studies (Table 2).

The MANOVA based estimated marginal means showed that all significantly different adaptation strategies except for reducing stocking density, the highland has the higher mean scores than the *kolla*. However, reducing stocking density in livestock has been used in the *kolla* than highlands. Estimated marginal means for significantly variate coping strategies show the higher mean score for renting outland in the highland; while the mean scores for change in consumptions, selling livestock, and remittance was higher in the *kolla* (Table 2ᵈ).

## Constraints to adaptation: The causal linkages along with case studies

The results of the focused group discussions revealed that the local community's livelihoods depended heavily on exploiting available natural resources, expanding human settlements, and

cultivating subsistence crops. The exploitations of these resources were free from central planning and interventions, but the profound imprints of nature on the local community. Thus, the adaptation and coping strategies were free, provided by the surrounding physical environments, and affected by the natural processes. The group discussants claim that the local community has had the experiences to change the cultivation methods, farming systems, and resource exploitations for their livelihood guided by the natural processes for the changes in the provisions of the local biophysical environments previously. However, socioeconomic situations and biophysical environment have been changing from time to time that most of these strategies become unpractical; and the local community has been enforced for a frequent change in most of their inherent adaptation and coping strategies. The discussants have attributed to different factors for the hurdles to the traditional strategies, rather than positive transformation. These factors varied among the FGD groups and across case studies.

Most of the adaptation and coping strategies experienced in the watershed in the past were associated with climate and environmental dynamics and the socioeconomic factors mainly resulted from demographic pressure. Thus, the exploiting natural forest for food and materials traveling long distance, seasonal migration in search of grass and water for cattle('*darabaa*' system) from highland to the *kolla* area, planting trees for economic and non-economic benefits, cultivations of early mature crops such as barley, beans, peas and cabbages which they called as '*dafoo*' and '*birroo*', mulching and fallowing to sustain soil fertility, and diversification of incomes through forest gathering and non-timber harvest such as honey productions were recognized in the local community for their previously practiced adaptation strategies. However, most of these strategies become less practical and have been replaced by the intensification of agriculture through improved varieties and artificial fertilizers, reducing stocking density, selling livestock, practicing irrigation to cultivate high-value crops such as fruits and vegetables for markets.

The group discussants explained that now a day, different coping strategies less known in the past have been adopting by the local community in the watershed mainly in the *kolla*. Some of these strategies include seasonal migration, selling labour for a wage, reducing consumption and changing dietary, and enhancing remittance through sending children/daughters abroad. These newly emerged strategies were explained as unpleasant systems, forced by situations, and perceived as livelihood and social risks among the FGD discussants mainly in the *kolla*. For example, the deteriorating agricultural extensification such as mulching, fallowing, slash and burn practices have been affecting the local community's stewardship for the land and the appreciation of land values. According to these discussants, livestock rearing was the crucial livelihood system in the *kolla*, and the current land-use and grazing situations that have been enforcing the local community to adopt the strategies such as the selling of livestock and reducing stocking would be resulted in perpetuating poverty in the area. Reducing food consumption and changing diets have destructive health problems and erodes the cultural values of the households.

Most of the newly adopted strategies which were negatively explained by the *kolla* discussants were similarly perceived in the highland, but the causalities were different along with the case studies. The highland discussants have remarkably associated the problems with the local biophysical and socioeconomic processes. Among recently adopted strategies, the agricultural intensification was focused during the discussions. Accordingly, this strategy became increasingly adopted by the local community because of two reasons: First, the need for the increased production from the land which became scarce. Second, to cover the household's food demand through these techniques because the livelihood options have been decreased mainly due to lack of access to the natural resources for the livelihood assets. However, this strategy, for instance, has resulted in reduced crop diversity, economically inefficient due to the need for

external inputs, and less adaptable among the community in its dietary and food preparation culture. Moreover, the discussants claimed that the increasing climate variability conditions have been affecting the strategies used to adapt/and cope with the seasonal food insecurity. For example, the local agroecological feasibility for the productions of 'dafoo' and 'birroo' crops has been reduced because of the late-onset and early cessation of rain. The 'dafoo' crops are named after their production characteristics within a short period by using the rain in the early rainy season known as 'belg'. These crops are crucial for the food security of rural poor in the next season of heavy rainfall known as 'kiremt' when food scarcity reaches its peak in most of the study area. Although the productions of 'birroo' crops take place on the off-set of the rainfall, the time of plenty of food, the crops in this season are mostly pulses which have significant contributions to the food culture and provide the sources of income for the community. The transhumans (the 'darabaa' system) have a significant adaptation role for the highland community, when the grassing become scarce in their locality and they migrate to the *kolla* areas for the grazing and water for their cattle, in the past. However, this is not practical today that the land-use changes in the *kolla* have hindered the economic and ecological benefits they obtained from this agroecology.

Although causes for the unpleasant strategies in the highland were attributed to the resource's shortage due to natural increase in human population and climate variability, the *kolla* discussant claimed that the new adaptation represents the impacts of historical interventions by the national/ and global change processes. The adverse of these processes on the community's access to land resources resulted in the distorted agroecological contexts of adaptation and coping strategies in the *kolla*. Accordingly, as far as the livelihood in the kolla that depend on the common-pool resources are concerned, the land deal processes and interventions for large-scale agricultural investments at the expenses of these resources have been hindered the sustainable planning and implementations of inherent local adaptation strategies. Thus, most of the emerging strategies are generated from socioeconomic and environmental complexities resulted from these structural land-use dynamics. They claim that most of the present-day adaptation challenges facing small-scale farmers in the *kolla* were the result of the development policies that unrecognized the local community's livelihood and adaptation contexts.

The conditions of the adaptive learning process among the local community were one of the important issues raised during focus group discussions. The highland discussants explained that the local community in their area has been learning from the changing situations, mainly climate characteristics, landholding size, and environmental carrying capacity. Accordingly, the present strategies adopted in the highland were perceived as the result of local dynamics and the local community have been handling according to the demands of these situations. However, the *kolla d*iscussant explained that all the changes taking place since the 1970s are 'catastrophic' that they did not understand the natural state of change, unlike the highlanders. The *kolla* discussants suspected their extant knowledge on how to handle the environments if the investors left the valley 'supposed new policy doing so happen' because the inherent characteristics of the agroecology such as natural endowment have been changed at an alarming rate, and the change courses are complex, rapid and unnatural, and the community has been rarely learning from the processes of these changes.

## Discussions

All human societies are adaptive to changes [1, 43], that the small-scale farmers in the Anger watershed are not exceptional. The local community in this area has had a long history of interactions with natural environments [44], with rich experiences of exploiting the natural

environment for the resources and knowledge to sustain their adaptation strategies to the impacts of changing environments. However, the recent global environmental changes and associated complexities have been shaping the local adaptation and coping strategies. These processes have been constraining the inherent local adaptation strategies that these conditions have been increasing from time to time with remarkable spatial differences in affecting the local adaptation. Thus, the geography of adaptation strategies becomes varied within a region. This study was intended to assess the local situations of adaptation and coping strategies in the Anger watershed by considering the spatio-temporal aspects of these processes at the local level. These were important to identify the local level status and trends in the adoptions of these strategies, the spatial conditions of adaptation sustainability, and constraints causalities in the local area through comparative case study technique.

Agroecology-based research on adaptation strategies in Ethiopia came up with several conclusions regarding the differences in types of mechanisms applied by farmers [45, 46], the determinants of adoptions [47, 48], and the adaptation constraints [49]. However, this study is different from the previous researches in its conceptual and methodology that potentially contribute to future research and policy recommendations on rural adaptation. First, we have considered the local contexts on the specific conditions non-climate change processes such as the cases differences on the extents of structural land use dynamics for their effects on the local adaptation strategies, and the biophysical situations of agroecology [28, 50]. Second, even those previous researches that considered agroecology, conceptualized for its production functionality, evaluated the status of the adaptation strategies by associating with climate variability situations along with these spatial differences. The climate change situations were the only considered external disturbance resulted from agroecology differences in these studies [51]. However, the reality in our study area revealed that agroecology referring to livelihood systems and adaptation strategies are more socioeconomic and political precinct than its ecological functionality. Thus, the adaptation status and trends, the sustainability of local adaptation strategies, and the constraints to adaptation strategies require a detailed understanding of local situations. Therefore, this broader conceptualization and comprehensive methodology applied in this study provide to capture the local contexts which were crucial to understand the sustainability conditions of the local adaptations and factors of adaptation constraints.

The observed differences in household adaptation situations along with the agroecology were different from the research results in the other parts of Ethiopia. Most of the previous studies revealed that the adaptation strategies in the *kolla* agroecology were relatively sustainable that the communities are more adaptive than their counter highlands [46, 48, 52] unlike the condition observed in the Anger watershed, where the local adaptation situations were more sustainable in the highland. Moreover, the types of adaptation strategies implemented were different from our cases. For example, reducing stocking density and seasonal livestock selling were identified as the strategies more adopted in the highland than *kolla* agroecology in the other parts of Ethiopia. These were inverse in our cases that the first strategy was equivalent, and the second was higher in the *kolla* of our study area.

The local community's access to the land resources has been cognized for the effects on the local adaptation strategies in the previous studies in Ethiopia. However, these studies were insufficient on the causalities that the local adaptation problems were mostly attributed to the biophysical dynamics resulted from climate variability and the socioeconomic situations internal to the community. However, this study signifies causal linkages between the effects of local processes of non-climatic environmental changes external to the local community and the sustainability situations of the local adaptation. This study revealed that the conceptual and methodological efforts for local adaptation studies should be toward understanding the local contexts of the livelihood disturbances and the effects of these local processes in constraining

the adaptation. To these ends, the in-depth explorations of the local community's perceptions on the causalities, how they perceive the relationship between the strategies they have been adopting, and the local processes of environmental changes were crucial. In this regard, this study revealed that the site-specific non-climate factors are significant to shape the community's adoptions on the types of adaptation strategies, and for their effects on the persistence of maladaptation, and in constraining the adaptation strategies labelled for their efficiency at the local level.

The national/ and global economic interventions were agroecological in the Anger watershed, and the impacts of the processes overweighted the power of ecological functionality. For instance, albeit livestock rearing is the most sustainable livelihood system of the *kolla* agroecology in Ethiopia, the strategies became less practical in the *kolla*. This condition was real in the study area before these interventions, when the highland community uses the *kolla* for their seasonal transhumans. Thus, opposite to the agroecological feasibility of the *kolla*, the farmers have been adopting reducing stock density and seasonal livestock selling. These adaptation strategies have been practiced in response to the effects of denied access to land resources resulted from large-scale agricultural investments. However, similar to other parts of Ethiopia, livestock rearing became less practical in the highland parts of the watershed due to biophysical, socioeconomic, and demographic factors, distinguished as internal to the local community.

As a typical ecological simplification of the state in rural Ethiopia [19], the interventions for 'development' in the *kolla* agroecology for the large-scale agricultural investments since the 1970s have been altered the local community's adaptation and coping strategies. to previously inaccessible and uninhabited areas served for free adaptation practices. These national/ and global economy induced 'development' policy [53, 54] in general and the state aspirations for the large scale agricultural [21] in particular, lends itself to be the key driver that complicates the rural adaptation strategies, as it is pointing up in the forced and maladaptation context of the farmers in the kolla case study; and these processes ultimately harm the national sustainable development. The disclosed current situation makes it hard for small-scale farmers in the kolla to reverse to the original state of adaptation strategies [55], but forcibly adapt to 'development' induced stressors in the phase of changing climate.

The causal linkage on adaptation constraints along case studies could be associated with the community's adaptive learning process [56]. The effects of stressors on the local livelihood systems have been affecting the adaptive learning of the *kolla* community compared to the highland mainly in two ways. First, these disturbances have interactive processes that resulted in the complexities of the impacts which have the power to hinder their adaptive learning [43] The effectiveness of the adaptive learning in the kolla was affected by the magnitudes and frequency of the stressors' interruptions, where the effects of structural change in land use and tenure insecurity were high due to frequent interventions [57]. Although learning from historical experiences is crucial to deal with stressor/s, the more the stressors are complex and frequent the less effective adaptive learning. The small-scale farmers' interactions with climate variability and internal dynamics in the socioeconomic and biophysical environment in the highland involved as an adaptive learning process, drawn from experience knowledge through gradual and stable changes helped them to moderate changes in the biophysical environments and changes in society in a sustainable manner, while these became less efficient in the *kolla*.

The local livelihood adaptation strategies insightful to ecological knowledge in the *kolla* have at a loss because of the transformations of the rural landscape, driven once by the socialists and at another time by the capitalist development policies. These processes result in the critical trade-offs in the study area that the land resources belong to the local community versus large scale agriculture is decisive to achieve national food security and enhance agricultural

exports. The empirical evidence from different parts of Ethiopia [19, 20, 53] and experiences from this study area revealed that all the time the development endeavours oriented by the large-scale agricultural investments were failed to meet the goals of these programs/projects and simultaneously has exposed substantial complexities to the local adaptation and coping strategies.

## Conclusions

The local adaptation and coping strategies of small-scale farmers in the Anger watershed were assessed through comparative case studies. The variances in the households' adoptions have resulted from their agroecology positions. The highest variations were explained by adaptation strategies in which highland was higher in these strategies than the *kolla* community, while the difference in agroecology has the lowest effects for most of the coping strategies, even though the adoptions for these strategies was higher in the *kolla* in this regard. Although adaptation persisted in all history of the community, free adaptation and coping strategies profoundly depend on the natural environments in the past have become unpractical now a day in the study area. Constraints to the adaptation in the watershed show remarkable differences along with case-studies for their causalities and magnitudes.

The impacts of climate variability and socioeconomic factors intentioned as internal to the local community were remarkable causes constraining the local adaptation in the highland, while the present-day adaptation and coping strategies in the *kolla* are forced by the historical state interventions of the local land resources for the expansions of large-scale development projects. These processes disrupted the free adaptation practices and eventually stimulated the reframing of the local adaptation and coping strategies to the impacts of large-scale agriculture on top of, and even rather than climate variability. The adaptive learning practices in the highland contribute to the continual changes from free adaptation strategies when resources were relatively abundant and climate was relatively less variable to several new adaptation strategies fit with existing resources and environmental situations. However, the small-scale farmers in the *kolla* forcefully negotiating with the strategies they believed as maladaptation. The study revealed that the rich experiences in the human–environment co-existence of local farmers insightful to ecological knowledge are aggravated by the transformation of the rural landscape to stimulated large-scale agricultural investments. Thus, broad-spectrum policy and practices which recognize the adverse impacts of site-specific constraints and their implication on the small-scale farmers' adaptation options are recommendable to stipulating the adaptation options of the community.

This study signifies the local agroecology in the Anger watershed for its more socioeconomic and political precincts than ecological functionality. The national/and global actors external to the local community have been influencing the local community's inherent adaptation strategies even shape the natural functions of ecologies than the climate variability. This study contributed to research and policy on rural adaptation strategies. Future studies interested in appraising local adaptation strategies along agroecology and any development planning in areas like the Anger watershed ought to consider the following issues: First, local contexts of livelihood systems, and impacts of external processes such as climate variability and non-climate national/and global interventions on the local adaptation. Second, the concept of 'agroecology of adaptation strategies' has paramount importance in considering the simultaneous impacts of locally bolded processes because the adaptation situations have resulted from the combined effects of these processes. Thirdly, a much stronger emphasis should be given to the non-production aspects of agroecology for the analysis of local adaptation situations. Lastly, understanding adaptation situations helpful in providing locally

representative recommendations requires comprehending these issues through wide-ranging research methods similar to the techniques applied in this study.

## Supporting information

**S1 Text. English language survey questionnaire developed to collect data from small-scale farmers household heads to appraise local adaptation and coping strategies to global environmental changes across agroecology in Southwestern Ethiopia.**
(DOCX)

**S2 Text. English language focused group discussion protocol developed to collect data from elderly small-scale farmers to appraise local adaptation and coping strategies to global environmental changes across agroecology in Southwestern Ethiopia.**
(DOCX)

## Acknowledgments

This work was supported by Addis Ababa University, Ethiopia. We thank the university for providing financial support for the first author to undertake this study. We also thank zonal, districts, and *kebeles* administrators for their cooperation during data collection. We would like to appreciate all of the participants in this study for their time and patience in responding to our interview questionnaire and discussions in focused groups.

## Author Contributions

**Conceptualization:** Gemechu Y. Ofgeha, Muluneh W. Abshare.

**Data curation:** Gemechu Y. Ofgeha.

**Formal analysis:** Gemechu Y. Ofgeha, Muluneh W. Abshare.

**Funding acquisition:** Gemechu Y. Ofgeha.

**Investigation:** Gemechu Y. Ofgeha, Muluneh W. Abshare.

**Methodology:** Gemechu Y. Ofgeha, Muluneh W. Abshare.

**Resources:** Gemechu Y. Ofgeha.

**Software:** Gemechu Y. Ofgeha.

**Supervision:** Muluneh W. Abshare.

**Validation:** Gemechu Y. Ofgeha, Muluneh W. Abshare.

**Visualization:** Gemechu Y. Ofgeha, Muluneh W. Abshare.

**Writing – original draft:** Gemechu Y. Ofgeha.

**Writing – review & editing:** Gemechu Y. Ofgeha, Muluneh W. Abshare.

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
