## [Decision Letter · Decision Letter 0]

7 Jun 2021

PONE-D-21-14280

Appraising local adaptation and coping strategies to global environmental changes: Portraying agroecology beyond production functions in Southwestern Ethiopia

PLOS ONE

Dear Dr. Gemechu,

Thank you for submitting your manuscript to PLOS ONE. After careful consideration, we feel that it has merit but does not fully meet PLOS ONE’s publication criteria as it currently stands. Therefore, we invite you to submit a revised version of the manuscript that addresses the points raised during the review process.

We look forward to receiving your revised manuscript.

Kind regards,

Shah Md Atiqul Haq

Academic Editor

PLOS ONE

Journal Requirements:

Additional Editor Comments:

Dear authors,

I would suggest that you follow the reviwers' comments and suggestions.

The paper has many weaknesses including results, discussions, formatting and English related.

I would ask you to revise the paper carefully and send it again for further consideration.

Yours sincerely,

Reviewers' comments:

Reviewer's Responses to Questions

**Comments to the Author**

1. Is the manuscript technically sound, and do the data support the conclusions?

Reviewer #1: Yes

Reviewer #2: Yes

Reviewer #3: Partly

2. Has the statistical analysis been performed appropriately and rigorously? 

Reviewer #1: Yes

Reviewer #2: Yes

Reviewer #3: Yes

3. Have the authors made all data underlying the findings in their manuscript fully available?

Reviewer #1: Yes

Reviewer #2: Yes

Reviewer #3: Yes

4. Is the manuscript presented in an intelligible fashion and written in standard English?

Reviewer #1: Yes

Reviewer #2: Yes

Reviewer #3: No

5. Review Comments to the Author

Reviewer #1: The manuscript is well written and justified with reference to previous studies. The work is novel and nicely compiled.

However few minor corrections are required before further processing.

Abstract is way too long, there is a need to concise it according to the journal requirements. Formatting of all the sections should be rechecked.

Headings should be according to the journal's format.

References need to be uniform in formatting.

Conclusion of the study should not be more then a paragraph or two. It should only focus on main outcome of the study. Conclusion section should be shortened as well.

Best wishes.

Reviewer #2: Thoroughly check the manuscripts and remove all the spelling mistakes. On the other hand, cross check all the references that quoted in the text with reference list. Discussion section needs improvements.

Line No : 709 correct the spelling (Acknowledgements)

Line No: 716 Replace Reference with References

Reviewer #3: The article need to improve according to format

and author should clear the results and discussion regarding the statistical analysis

There should evidence from where collect the data

Should mention related references

Grammatically mistakes should be removed

6. PLOS authors have the option to publish the peer review history of their article (what does this mean?). If published, this will include your full peer review and any attached files.

Reviewer #1: **Yes: **Dr. Adnan Akhter

Reviewer #2: No

Reviewer #3: No

---

## [Author Response · Author response to Decision Letter 0]

1 Jul 2021

Editor Comments: Dear authors, I would suggest that you follow the reviewers' comments and suggestions. The paper has many weaknesses including results, discussions, formatting and English related.

(Response): We appreciate you and the reviewers for your precious time in reviewing our paper and providing valuable comments. We have carefully considered the comments, incorporated the suggestions, and tried our best to address every one of them. We hope the manuscript after careful revisions meet your high standards. Specifically, we have carefully considered removing the grammatical problems in this manuscript, and an exhaustive correction was made in this revised version. However, these grammatical corrections were too much that we didn’t include in this response letter, and in the ‘Manuscript Revised with highlighted change’

Reviewer #1

(Comment: General) The manuscript is well written and justified with reference to previous studies. The work is novel and nicely compiled. However, few minor corrections are required before further processing.

(Response): Thank you very much. We have incorporated the suggested correction in all parts of this paper.

(Comment: P2: L26-58) “Abstract is way too long, there is a need to concise it according to the journal requirements.”

(Response): Thank you very much. We have revised the abstract based on your comments.

(Comment: General) “Formatting of all the sections should be rechecked.”

(Response): Thank you. We have rechecked all sections of the paper based on the journal’s format. 

(Comment: General) “Headings should be according to the journal's format.”

(Response): Thank you for the suggestions. We have made revisions for the heading and subheadings as follows: 

• Conceptual frameworks (page 5: Line 137)

• Research methods and materials (page 8: Line 238)

(Comment: P27: L840-1009) “References need to be uniform in formatting.”

(Response): Thank you very much. We have checked the references for its consistency. 

(Comment: P24: L742) “Conclusion of the study should not be more than a paragraph or two. It should only focus on main outcome of the study”

(Response): Thank you for the suggestions. We have tried to condense the conclusions but it is difficult to conclude for all specific objectives within one or two paragraphs.

Reviewer # 2

(Comment: General) “Thoroughly check the manuscripts and remove all the spelling mistakes.”

(Response): Thank you for the suggestions. We have made revisions accordingly.

(Comment: P27: L840-1009) “cross check all the references that quoted in the text with reference list.”

(Response): Thank you very much. We have rechecked all the citations in the texts against the reference list. We found that the citation on the Page 13: Line 399 and page 17: Line 490 was not referenced. Now it is included in the references list as follows: 

39. Mbang A, Owino J, Kones J, Meffeja F, Dibog L. Evaluation of Ensiled Brewer’ s Grain in the Diet of Piglets by One Way Multiple Analysis of Variance, MANOVA. 2007. 

41. Pallant J. Survival Manual:A step by step guide to data analysis using SPSS. 4th ed. New York, USA: Open University Press; 2010. 

(Comment: P21: L614) “Discussion section needs improvements.”

(Response): Thank you for the comments. We have made revisions that hopefully this part after careful revisions meet the required standard.

(Comment: P27:L833) “Line No: 709 correct the spelling (Acknowledgements)”

(Response): Thank you very for the suggestion. We have revised. 

(Comment: P27: L840) “Line No: 716 Replace Reference with References”

(Response): Thank you for the suggestion. We have revised. 

Reviewer #3

(Comment: General) “The article needs to improve according to format”

Response: Thank you very much for the suggestion. We have revised the manuscript according to the journal’s format. Several formatting errors such as font size, text formatting, and the heading and subheadings were corrected in this revised version. 

(Comment: General) “author should clear the results and discussion regarding the statistical analysis.”

(Response): Thank you very much. We have carefully considered the statistical assumptions and rules in the analysis applied in this study. It is our pleasure to accept you, in case if you have any specific comments/ suggestions regarding the statistical analysis. 

(Comment: P11: L337-339) “There should evidence from where collect the data. Should mention related references”

(Response): Thank you very much for the comments. We have accepted all the comments, and rewritten this part to be clear with the sources of data as follows: The small-scale farmers’ household heads, and the focus group discussants drawn from local notoriety individuals and the elderly in the Anger watershed, and experts from relevant zonal and districts offices in the east Wallagga administrative zone were the primary sources of data (P11: L337-339)

(Comment: General) “Grammatically mistakes should be removed”

(Response): Thank you very much. We have carefully considered the comments regarding the grammatic problems in this manuscript, and an exhaustive correction was made in this revised version. However, these Grammarly corrections were too much to include in this response letter.

---

## [Decision Letter · Decision Letter 1]

21 Jul 2021

PONE-D-21-14280R1

Local adaptation and coping strategies to global environmental changes: portraying agroecology beyond production functions in southwestern Ethiopia

PLOS ONE

Dear Dr. Gemechu Yigezu Ofgeha,

Thank you for submitting your manuscript to PLOS ONE. After careful consideration, we feel that it has merit but does not fully meet PLOS ONE’s publication criteria as it currently stands. Therefore, we invite you to submit a revised version of the manuscript that addresses the points raised during the review process.

We look forward to receiving your revised manuscript.

Kind regards,

Shah Md Atiqul Haq

Academic Editor

PLOS ONE

Journal Requirements:

Additional Editor Comments (if provided):

Dear authors,

Please address the final remarks of the reviwers.

I am looking forward to receiving the final revised version.

Reviewers' comments:

Reviewer's Responses to Questions

**Comments to the Author**

1. If the authors have adequately addressed your comments raised in a previous round of review and you feel that this manuscript is now acceptable for publication, you may indicate that here to bypass the “Comments to the Author” section, enter your conflict of interest statement in the “Confidential to Editor” section, and submit your "Accept" recommendation.

Reviewer #1: All comments have been addressed

Reviewer #2: All comments have been addressed

2. Is the manuscript technically sound, and do the data support the conclusions?

Reviewer #1: Yes

Reviewer #2: Yes

3. Has the statistical analysis been performed appropriately and rigorously? 

Reviewer #1: Yes

Reviewer #2: Yes

4. Have the authors made all data underlying the findings in their manuscript fully available?

Reviewer #1: (No Response)

Reviewer #2: Yes

5. Is the manuscript presented in an intelligible fashion and written in standard English?

Reviewer #1: Yes

Reviewer #2: Yes

6. Review Comments to the Author

Reviewer #1: (No Response)

Reviewer #2: Comments for authors

i) Conclusion section is too lengthy please try to shorten it if possible

ii) Carefully cross check all the references that quoted in the text with reference list

iii) Over all article is suitable/fit for publication in Plos One Journal

7. PLOS authors have the option to publish the peer review history of their article (what does this mean?). If published, this will include your full peer review and any attached files.

Reviewer #1: **Yes: **Dr. Adnan Akhter

Reviewer #2: No

---

## [Author Response · Author response to Decision Letter 1]

23 Jul 2021

Once again, we appreciate the editor and reviewer for their precious time in reviewing our paper and providing valuable comments. We have carefully considered the comments, incorporated the suggestions, and tried our best to address every one of them. We hope the manuscript after careful revisions meet your high standards. 

Our responses hereunder consist the comments regarding the journal requirements and the editor’s and reviewers’ comments on the manuscript. 

(Comments on the Journal Requirements): Please review your reference list to ensure that it is complete and correct. If you have cited papers that have been retracted, please include the rationale for doing so in the manuscript text, or remove these references and replace them with relevant current references. Any changes to the reference list should be mentioned in the rebuttal letter that accompanies your revised manuscript. If you need to cite a retracted article, indicate the article’s retracted status in the References list and also include a citation and full reference for the retraction notice.

(Response): Thank you for the suggestion. We have carefully considered regarding the references of this manuscript. Now, we have ensured that it is complete and correct. Moreover, the issue of retracted articles is fully considered in this revision. Thus, we approved that no retracted sources were referenced in this manuscript. 

(Editor Comments): Dear authors, please address the final remarks of the reviewers.

(Response): We appreciate you and the reviewers for your precious time in reviewing our paper and providing valuable comments. We have carefully considered the comments, incorporated the suggestions, and tried our best to address every one of them. We hope the manuscript after careful revisions meet your high standards. 

Reviewer #1

No comment provided

Reviewer # 2

(Comment: General) “Conclusion section is too lengthy please try to shorten it if possible.”

(Response): Thank you for the suggestions. We have tried to condense the conclusions again. However, our concern is that trying to shorten more will result in missing to conclude for all specific objectives of the study. 

(Comment: General) “Carefully cross check all the references that quoted in the text with reference list.”

(Response): Thank you very much for the reminder. We have carefully rechecked all the citations in the texts against the reference list. 

(Comment: General) “Over all article is suitable/fit for publication in Plos One Journal.”

(Response): Thank you very much for the recommendations. Yours and other reviewers’ valuable comments have pivotal role for the fitness of this manuscript.

---

## [Editor Report · Decision Letter 2]

26 Jul 2021

Local adaptation and coping strategies to global environmental changes: portraying agroecology beyond production functions in southwestern Ethiopia

PONE-D-21-14280R2

Dear Gemechu Yigezu Ofgeha,

We’re pleased to inform you that your manuscript has been judged scientifically suitable for publication and will be formally accepted for publication once it meets all outstanding technical requirements.

Kind regards,

Shah Md Atiqul Haq

Academic Editor

PLOS ONE

Additional Editor Comments (optional):

Dear authors,

Congratulations!!!

Your paper has been accepted.
---

## [Editor Report · Acceptance letter]

29 Jul 2021

PONE-D-21-14280R2 

Local adaptation and coping strategies to global environmental changes: portraying agroecology beyond production functions in southwestern Ethiopia 

Dear Dr. Ofgeha:

I'm pleased to inform you that your manuscript has been deemed suitable for publication in PLOS ONE. Congratulations! Your manuscript is now with our production department. 

Kind regards, 

on behalf of

Dr. Shah Md Atiqul Haq 

Academic Editor

PLOS ONE